

# 1 Widespread increase of root zone storage
# 2 capacity in the United States

Jiaxing Liang[1], Hongkai Gao[1]*, Fabrizio Fenicia[2], Qiaojuan Xi[1], Yahui Wang[1],
Hubert H. G. Savenije[3]
[1] School of Geographic Sciences, East China Normal University, Shanghai,
China
[2] Eawag, Swiss Federal Institute of Aquatic Science and Technology, Dubendorf,
Switzerland
[3] Water Resources Section, Delft University of Technology, Delft, the
Netherlands
*Corresponding author. Email: hkgao@geo.ecnu.edu.cn
https://orcid.org/0000-0003-0786-8067

## 15 Abstract

The root zone is the upper part of the unsaturated zone, where water and
nutrients are accessible to plants, controlling hydrological responses, vegetation
dynamics, biogeochemical processes, and land-atmospheric interaction. The root
zone storage capacity ($S_{umax}$) represents the maximum subsurface moisture
volume that can be accessed by the vegetation's roots, controlling the
partitioning of precipitation into storage, runoff and percolation. Previous work
has illustrated that $S_{umax}$ varies spatially, largely responding to climatic
conditions. It can be therefore expected that $S_{umax}$ varies temporally as well in
response to climate change. However, this hypothesis has not been tested. In this
study, we utilized a conceptual hydrological model and a dynamic parameter
identification analysis method, to quantify the temporal trends of $S_{umax}$ for 497
catchments in the USA. We found that 423 catchments (85%) showed increasing
$S_{umax}$, which averagely increased from 178 to 235 mm between 1980 and 2014.
The increasing trend was also validated by multi-sources data and independent
methods. Our results suggest that ecosystems dynamically adapt their root zone
in response to climate change, which significantly affects hydrological processes
and water resources availability. Moreover, the increase of $S_{umax}$ significantly
correlates to hydroclimatic indicators and vegetation dynamics. These results
highlight the importance of considering the co-evolution of climate, ecosystems,
and hydrology.



## 1. Introduction

The root zone water storage capacity ($S_{umax}$) is the size of a conceptualized bucket in the unsaturated zone of the soil in which vegetation buffers water during wet periods to sustain transpiration during dry periods (Gao et al., 2014a). The real shape of the $S_{umax}$ is hard to determine, as it consists of a complex of pores and fissures in the substrate and extends both laterally and in depth. Generally, this volume is normalized by the area, and it is therefore represented as a depth. The $S_{umax}$ forms a crucial link between ecosystems and hydrological processes (Dralle et al., 2018; de Boer-Euser et al., 2019a; Gao et al., 2023). It controls the partitioning of precipitation in flow generation and plant use. It forms a core parameter in conceptual hydrological models (Fenicia et al., 2011; Seibert and Vis, 2012; Zhao, 1992; Gao et al., 2023). The accurate estimation of $S_{umax}$ is essential for global and regional hydrological simulation, land surface processes, and dynamic vegetation modeling.

Observing $S_{umax}$ directly is impractical. Traditional methods use field measurements of rooting depth and soil texture to estimate plot scale $S_{umax}$, under the assumption that soil properties determine plant available moisture (Jackson et al., 1996;Schenk and Jackson, 2002). However, this method is labor intensive, costly and destructive. Moreover, it only provides local and sparse estimates with large uncertainty of upscaling. More importantly, this method provides instantaneous measurements, which cannot reflect the dynamic response of the root zone to climate change or human activities.

An inverse approach to determine $S_{umax}$ is to look at ecosystem performance and what it does to buffer against dry spells. The water-balance based Mass Curve Technique (MCT) provides a powerful tool to derive the root zone storage capacity by observable land surface moisture fluxes, including precipitation, snowmelt, evaporation, runoff, and human-induced irrigation (Gao et al., 2014a; Wang-Erlandsson et al., 2016). Another inverse approach to determine $S_{umax}$ is by parameter calibration (Fenicia et al., 2008a; Gharari et al., 2014; Merz et al., 2011), which can serve as a benchmark in well-gauged catchment-scale studies. This method also has uncertainties, mostly associated to model parameter equifinality and the difficulty of relating model parameters to catchment characteristics. However, such problems can be attenuated through specific modelling choices, and this method can provide useful indications of otherwise unobservable properties.

It is well documented that, because of climate change and land-use management, ecosystems have adjusted their above-ground biomass, leading to a greening trend at global-scale (Lele and Krishnaswamy, 2019; Chen et al., 2019). However, little is known about terrestrial ecosystems' root zone adaption to these changes. To explore the root zone dynamics, we used multi-source datasets to determine temporal changes of $S_{umax}$ and compared them to other environmental indicators.

In this study, we utilized a large-scale catchment dataset of 497 catchments in the USA. We used two independent approaches for $S_{umax}$ estimation. The first



one uses a parsimonious conceptual hydrological model (FLEX), which is
calibrated in a moving time window, using the Dynamic Identification Analysis
method (DYNIA) approach (Wagener et al., 2003). The second approach is based
on the MCT method, with the ERA-5 reanalysis grid-cell data as forcing, which
provides the root zone storage capacity in different ecosystems required to
overcome certain return periods of droughts, i.e. 5, 10, 20, 30, 40 years. We
compared the values and trends of root zone storage capacities from these two
independent methods and analyzed the temporal trends of $S_{\text{umax}}$ in relation to
environmental variables.

## 90  2. Data and Methods

### 91  2.1 Data

The hydrometeorological data used in this study is the CAMELS
(Catchment Attributes and MEteorology for Large-sample Studies) dataset
(https://doi.org/10.5065/D6MW2F4D) (Addor et al., 2017). The CAMELS
dataset comprises daily meteorological data and catchment attributes from 1980
to 2014 for 671 catchments across the United States. It covers a wide variety of
hydroclimatic conditions, including long streamflow time series from
catchments with limited impact by human activities. Catchment-scale
precipitation and temperature were obtained from the Daymet data set (Thornton
et al., 2012). Potential evaporation was estimated based on temperature data,
using the Hargreaves equation (Hargreaves et al., 1985). The NDVI data is the
current release of the NOAA Global Inventory Monitoring and Modeling System
(GIMMS) long series (1981-2015) homogenized vegetation index product with
version number 3g.v1 (https://doi.org/10.3334/ORNLDAAC/2187) (NCAR,
2018; Pinzon and Tucker, 2014; Thornton et al., 2016).
The catchments with missing daily data were eliminated, and only those
with complete data were retained. This filtering resulted into 497 catchments
(Figure 1). The 497 catchments were classified into 10 clusters according to
Jehn et al. (2020), based on climate, hydrology and location. The 10 clusters are
determined using a principal component analysis based on Ward's linkage
method (Ward, 1963). Figure 1 and Table 1 present distribution maps and
detailed information for 10 clusters, while Figure 2 illustrates the temporal
trends of hydroclimatic variables and NDVI for these 10 clusters.
ERA5 reanalysis precipitation, evaporation, snowmelt and irrigation data
are also used as part of the methodology in order to provide an independent
validation (https://cds.climate.copernicus.eu/cdsapp#!/software/app-c3s-daily-
era5-statistics?tab=app). In particular, this data will be used as input for the
MCT method (see Section 2.4), to estimate the temporal variation of root zone
storage capacity.





### 2.2 Model calibration approach

#### 2.2.1 Model description

The first approach for estimating $S_{umax}$ is based on model calibration. The model used in this research is based on the FLEX hydrological model (Fenicia et al., 2009; Fenicia et al., 2011; Gao et al., 2014b). The model is composed of reservoirs, lag functions and junction elements to represent different hydrological functions constructed with the flexible modelling framework SUPERFLEX (Fenicia et al., 2011). It includes five reservoirs (Figure 1): a snow reservoir ($S_w$), an interception reservoir ($S_i$), a root zone reservoir ($S_u$), a fast-response reservoir ($S_f$) and a slow-response reservoir ($S_s$). The water budget equation and structural equation of different reservoirs are shown in Table 2. There are 10 free parameters that need to be calibrated, as shown in Table 3, which describes the role of each parameter and the bound of its value.

Precipitation is stored in snowpack or interception reservoirs before entering the root zone reservoir. Snow accumulation and melting are calculated based on a degree day factor algorithm. When the temperature is below the threshold temperature $T_t$ (°C), precipitation $P$ (mm day$^{-1}$) occurs as snowfall ($P_s$, mm day$^{-1}$), and increases the storage in snow reservoirs $S_w$. When the temperature is above the threshold temperature $T_t$, the amount of snow melting $M$ can be calculated from the parameter $F_{DD}$ (mm d$^{-1}$ ℃$^{-1}$) (Eq. (7)).

The precipitation retained in the interception reservoirs is directly returned to the atmosphere by evaporation. The interception evaporation $E_i$ (mm d$^{-1}$) is the same as potential evaporation $E_p$ (mm day$^{-1}$) if there is water in the reservoir. The storage volume in the interception reservoir is $S_i$ and the maximum storage capacity is $I_{max}$ (mm) (Eqs. (8) to (10)).

The core of this hydrological model is the root zone module, which determines the partitioning of effective precipitation ($P_e$) into either runoff generation ($R_u$) or evaporation ($E_a$).

The actual evaporation $E_a$ (mm d$^{-1}$) in the soil is determined by potential evaporation $E_p$, actual storage in root zone $S_u$, $S_{umax}$ (mm) and parameter $C_e$ (-) (Eq. (12)). Runoff generation ($R_u$) is determined by the amount of effective precipitation ($P_e$), the actual storage in root zone ($S_u$), and the root zone moisture storage capacity ($S_{umax}$). In equations (13) and (14), $C_r$ (-) represents the runoff coefficient, $\beta$ (-) is the spatial diversity factor, and $R_u$ (mm) represents the generated flow during rainfall events, obtained by multiplying the effective rainfall and snowmelt $P_e$ (mm) entering the soil module by the runoff coefficient $C_r$.

The generated runoff $R_u$ is divided through the parameter $D$ (-) into fast response runoff and slow response runoff (Eq. (15) and Eq. (16)). Equations (17) and (18) were used to describe the time lag between storm and fast runoff. $R_f$ (mm) is the generated fast runoff, $T_{lag}$ (d) is a parameter which represents the lag-time between storm and fast runoff generation, $c(i)$ is the weight of the flow in $i-1$ days before and $R_{fl}$ (mm) is the runoff into the fast-response reservoir $S_f$ after convolution. Slow runoff $R_s$ (mm) into the slow-response reservoir $S_s$.



A linear equation was used to conceptualize the flows in the fast response
reservoirs and slow response reservoirs. In equations (20) and (22), $S_f$ and $S_s$
represent the fast and the slow reservoirs; $K_f$ (d) and $K_s$ (d) represent the fast
and slow receding coefficient; $Q_f$ and $Q_s$ represent the fast and slow runoff,
respectively, while simulated runoff $Q_m$ is the sum of the $Q_f$ and $Q_s$.

**2.2.2 Dynamic parameter identification and model evaluation**

The assessment of the temporal variation of parameters is based on the
Dynamic Identification Analysis method (DYNIA) proposed by Wagener et al.
(2003). DYNIA is based on a Monte Carlo framework and employs a Latin
hypercube sampling technique. In this study, we generated 40,000 sets of
parameter combinations within the feasible range for the 10 parameters. Each
set of parameters is associated to a streamflow simulation, for which a
performance metric is calculated. In this study, the Kling-Gupta efficiency
(KGE) proposed by Gupta et al. (2009) and modified by Kling et al. (2012) was
used to calculate the model simulation performance. Model performance was
calculated using a five-year moving window, using the first year of each period
as a warm-up. For each period and each catchment, the optimal model (with
highest KGE) was selected. Subsequently, we considered the 10 catchments
clusters provided by Jehn et al. (2020), and averaged the optimal parameters for
all catchments in the same cluster and period. The simulation results of temporal
trends for all 10 parameters of 10 clusters are shown in Figure 4.

**2.3 MCT method**

As an alternative to model calibration, root zone storage capacity has been
determined through the mass balance method using merely climatic data. This
storage capacity is referred to as $S_R$, which is subsequently compared to the
model-derived $S_{umax}$ as an independent validation.

The MCT method estimates $S_R$ based on the principle of the water balance (Gao et al.,
2014a; Wang-Erlandsson et al., 2016). When the outflow ($F_{out}$) from the root zone (i.e.
evaporation) exceeds the inflow ($F_{in}$) (i.e. infiltration), then the water deficit is calculated by
computing the difference between the two. This water deficit requires plants to draw water
from storage ($S_R$). Vegetation's canopy interception ($I$) is also considered in the MCT method,
to make corresponding comparison with the calibrated FLEX.

$F_{in}$ represents the sum of net precipitation ($P$-$I$), snowmelt ($SM$), and irrigation ($IRR$):

$F_{in} = P - I + SM + IRR$    (18)

Since $E$ in ERA-5 data is the total evaporation, including canopy interception, the
evaporation from root zone ($F_{out}$) must subtract the amount of interception from the total
evaporation:

$F_{out} = E - I$  (19)

The difference between outflow and inflow then equals $P + SM + IRR$ - $E$, where the
interception drops out. This difference is accumulated on a daily scale:



$$A(t_\mathrm{n} \to t_\mathrm{n+1}) = \int_{t_\mathrm{n}}^{t_\mathrm{n+1}} F_\mathrm{out} - F_\mathrm{in} dt \quad (20)$$
Where $A(t_\mathrm{n} \to t_\mathrm{n+1})$ represents the water deficit on day $t_\mathrm{n+1}$. The sum of daily water deficits
constitutes the cumulative water demand:
$$D_e(t_\mathrm{n+1}) = \max(0, D_e(t_\mathrm{n}) + A(t_\mathrm{n} \to t_\mathrm{n+1}))(21)$$
$D_e(t_\mathrm{n+1})$ represents the cumulative water deficit on day $t_\mathrm{n+1}$. The accumulation of $D_e$ only
occurs during periods when $F_\mathrm{out} > F_\mathrm{in}$, while a reduction in $D_e$ occurs when $F_\mathrm{out} < F_\mathrm{in}$.
Additionally, $D_e$ has a minimum value of 0. The required root zone storage capacity $S_\mathrm{R}$
represents the maximum value of $D_e$:
$$S_\mathrm{R} = \max(D_e(t_0), D_e(t_1), D_e(t_2), \dots, D_e(t_\mathrm{end}),)(22)$$
To account for the impacts of multi-year droughts, we allow the deficit $D_\mathrm{e}$
accumulation continues in the end of the end, and extends into the following
year. Then the maximum $D_\mathrm{e}$ of that year is regarded as the year's $S_\mathrm{R}$. Since $S_\mathrm{umax}$
is simulated using a five-year time window, to make fair comparison, the
maximum $S_\mathrm{R}$ value over the same five-year period was compared with $S_\mathrm{umax}$.
Different ecosystems have different strategies to cope with drought. For
instance, forests, due to their longer lifetime, have a strong drought adaptation
requirement, resulting in a root zone storage capacity to overcome a drought that
may occur once in 20-40 years, while shrubs, having a shorter lifetime, exhibit
weaker adaptation demands, lasting through droughts occurring less frequently
than once every 20 years. Grasslands, on the other hand, can go dormant and
may accept a much higher probability of drought. Seasonal crops may permit a
probability of failure of once in 5 years. As a result, we calculated $S_\mathrm{R}$ for
different drought return periods ($S_\mathrm{R10y}$, $S_\mathrm{R20y}$ and $S_\mathrm{R40y}$) by applying the Gumbel
distribution to the yearly $S_\mathrm{R}$ (Gumbel, 1935). Many studies have shown that the
MCT method for estimating $S_\mathrm{R}$ is reliable (de Boer-Euser et al., 2016, 2019b;
Sakschewski et al., 2021; Wang et al., 2021; Wang-Erlandsson et al., 2016). The
MCT method utilizes the ERA-5 dataset, introduced in Section 2.1.

## 2.4 Correlation analysis

The Spearman correlation coefficient was utilized to quantify the
correspondence between temporal trends of $S_\mathrm{umax}$ and catchment environmental
changes. For each catchment, we calculated the time series correlation between
the catchment's precipitation $P$, runoff $Q$, temperature $T$, potential evaporation
$Ep$, runoff coefficient $Q/P$, evaporation coefficient $E/P=1-Q/P$ (assuming the
delta of water storage at annual scale is small), aridity index $AI$, precipitation
seasonality index $SI$, $NDVI$ and $S_\mathrm{umax}$. Each indicator is representative of a 5
years period, with 7 data points in each regression. Results are shown in section
241 3.3.



# 3. Results

## 3.1. Climate and environment changes

First we analyzed the changes in climate and vegetation data for the 497 study catchments from 1980 to 2014. We adopted the clusters provided by Jehn et al. (2020) to classify 497 catchments into 10 clusters, according to catchments characteristics in terms of climate, hydrology and location. These 10 clusters capture the unique hydrologic behavior of the continental United States and represent catchment groups with distinctly different hydrologic behavior. Figure 2 shows the spatially mean variations of precipitation $P$, runoff $Q$, temperature $T$, potential evaporation $Ep$, and $NDVI$ for the catchments in 10 clusters.

The average annual precipitation of clusters 3, 5, 6, and 7 generally exceed 1500 mm/yr, and clusters 1, 4, and 7 are the second largest with above 1000 mm/yr. Clusters 2, 8, and 9 are drier, with the lowest precipitation (<1000 mm/yr). The runoff characteristics of the catchments also reflect this precipitation pattern. From 1980 to 2014, all clusters experienced an upward trend in mean temperature, with cluster 3 showing the most significant increase of nearly 2°C. There were interannual variations in potential evaporation in 10 clusters, but no clear trends were observed. Except for clusters 3, the mean NDVI of the remaining clusters displayed an upward trend. There was a noticeable abrupt change occurring around 1990. Specifically, the most significant increase in NDVI took place before and after this time.

## 3.2. Spatial Patterns of $S_{umax}$ and $S_R$

We compared the $S_{umax}$ parameter of the FLEX model (representing the root zone storage capacity in catchment scale by parameter calibration) with the $S_R$ obtained from the MCT method (representing the root zone storage capacity in grid scale by land surface fluxes measurements, modeling and data assimilation), both exhibit similar spatial patterns in terms of magnitude and range (Figure 3).

Consistently with the predefined clusters, we found that catchments in the same cluster tend to behave similarly and catchments in different cluster can have different behavior. In particular, within clusters 1, 3, and 9, $S_{umax}$ exhibits the highest consistency with the 10-year drought return period ($S_{R10y}$) results. Clusters 4 and 7 are most aligned with $S_{R20y}$. Conversely, clusters 2, 5, 6, 8 and 10 are closest to $S_{R40y}$. The average root zone storage capacity for all catchments in the CAMELS dataset is most in line with the results for a 20-year drought return period ($S_{R20y}$).

Clusters 2 and 8 represent arid catchments with larger $S_{umax}$ values (>200mm), where vegetation often possesses deeper root systems to meet their water needs and avoid water stress. Cluster 9 is highly similar to Cluster 8 in terms of catchment characteristics but features higher forest coverage, with the



widest range in $S_{umax}$ distribution (200-300mm). The $S_{umax}$ values in Clusters 5,
6, and 7 are approximately 200mm. These catchments share similar
characteristics (Jehn et al., 2020) and are all located in the West Coast forest
region (Figure 1), known for abundant precipitation and strong seasonality.
Cluster 6 exhibits the most pronounced seasonality among all clusters, with the
majority of precipitation occurring in winter. By the end of summer, catchments
in this cluster are nearly completely dry. On the contrary, catchments in Clusters
3, 4, and 10, characterized by higher relative humidity and vegetation cover,
exhibit lower $S_{umax}$ values.

**3.3. Temporal variation of $S_{umax}$ and $S_R$**

The temporal variations of 10 the parameters of the FLEX model, calculated
using the DYNIA method, are shown in Figure 4. Except for the trend of $S_{umax}$,
there are some other interesting trends. For example, the threshold temperature,
$T_t$, controlling the split of snowfall and rainfall, dramatically increased in the
catchments of cluster 3, which have large amount snowpack. We believe it is
worthwhile to conduct further studies to understand the impacts of climate
change on this essential snow-related parameter. However, since this is out the
scope of this study, we did not implement detailed research, and focused this
study on the temporal change of $S_{umax}$.
The DYNIA results reveal that from 1980 to 2014, the annual average $S_{umax}$
for all 497 catchments increased from 178 mm to 235 mm, marking a 32%
increase, with a linear regression rate of 1.91 mm/yr (Figure 5). Across the 10
clusters, all $S_{umax}$ values exhibited an overall increasing trend. Specifically,
Clusters 1, 2, 9, and 10 showed noticeable upward trends, with Cluster 9
demonstrating the most significant increase, having a linear slope of 2.73
mm/yr. In contrast, Cluster 3 displayed the smallest growth in $S_{umax}$, with a slope
of only 0.03 mm/yr. Cluster 3 is characterized by a relatively small number of
catchments, only 6 in total, and is notable for its abundant snowfall (Jehn et al.,
2020). As shown in Figure 4, snow processes may play a more significant role
than the root zone in influencing $S_{umax}$. The increase of $S_{umax}$ suggests that the
ecosystems in these catchments adapted to environmental change by increasing
their root zone storage capacity (Dai, 2011; Gamelin et al., 2022).
The $S_R$ values obtained from the MCT method are highly comparable to
$S_{umax}$. The annual average $S_{R20y}$ for all 497 catchments also exhibited an
increasing trend, rising from 190 to 222 mm, with a linear regression rate of
1.07 mm/yr. From 1980 to 2014, $S_R$ increased by 32 mm, which is considerably
less than the increase in $S_{umax}$ derived from calibration. Among the 10 clusters, 8
clusters displayed an increasing trend in $S_R$, consistent with the trend in $S_{umax}$.
The only exceptions were Clusters 6 and 7, which showed decreasing trends
with $S_{R40y}$ and $S_{R20y}$ slopes of -0.74 mm/yr and -0.19 mm/yr, respectively. We
will discuss the possible reasons in the discussion.
Furthermore, from the perspective of individual catchments, $S_{umax}$ increased
in 85 % (423) of the catchments and decreased in 15 % (74) of the catchments
(Figure 6a). Catchments with increase in $S_{umax}$ were distributed throughout the
United States, while catchments with decrease in $S_{umax}$ were concentrated in the





western and central regions of the United States. This indicates that the
widespread increase of $S_{umax}$ occurs in most catchments in the United States.
Figure 6b demonstrated the comparison of the trends of $S_R$ and $S_{umax}$ in 497
catchments. Among these, 400 catchments exhibit an increasing trend in $S_R$,
while 97 catchments show a decreasing trend. In two-thirds (69%) of the
catchments, both $S_{umax}$ and $S_R$ display consistent increasing trends. Additionally,
17 catchments (3%) exhibit consistent decreasing trends in both $S_{umax}$ and $S_R$.
28% of the catchments demonstrated opposing trends between $S_{umax}$ and $S_R$.
Overall, root zone storage capacity ($S_{umax}$ and $S_R$) obtained using different
methods and different data exhibit similar trends and magnitudes in most
catchments (72%). The comparable results obtained by multi-sources datasets
and independent methods suggest that the trend changes in $S_{umax}$ do represent the
significant ecohydrological changes, rather than the result of parameter
uncertainties resulting from model calibration.
**3.4. Relationship between environmental change and $S_{umax}$ variation**
When comparing the variability of $S_{umax}$ with other indicators, it can be
seen that the temporal variation of $S_{umax}$ exhibits a positive correlation with $P$,
$T$, $Ep$, $E/P$, $SI$ and $NDVI$ in most catchments (Figure 7). The median correlation
coefficients range from 0.07 to 0.46. On the contrary, the temporal variation of
$S_{umax}$ is negatively correlated with $Q$, $Q/P$ and $AI$, and median range from -0.21
to -0.46. The temporal variation of $S_{umax}$ shows the strongest positive correlation
with $E/P$ (evaporation coefficient) and consequently the strongest negative
correlation with $Q/P$ (runoff coefficient), which in the long term equals 1-$E/P$.
The significant correlation of $S_{umax}$ with hydroclimatic indicators underscores
the interdependency of vegetation and hydrology, emphasizing the importance of
studying changes in root zone storage capacity for understanding hydrological
responses under changing conditions.
The correlations between environmental factors and $S_{umax}$ can vary
significantly among different clusters or catchments, even when the same
combination of factors is present (Figure 8). This variability can be interpreted
as arising from differences in catchment topography and hydrological processes.
For example, the results of our research demonstrate that $S_{umax}$ and $AI$ show a
negative temporal correlation in most catchments. Theoretically, the availability
of vegetation water is influenced by the humidity of the catchment, with larger
$S_{umax}$ observed in regions of higher aridity (Stocker et al., 2023). However, the
trend of $S_{umax}$ was negatively correlated with the $AI$ in most of the catchments in
the clusters (1, 3, 5, 6, 7, 10) in wetter regions (Figure. 8g). This may be
explained by the fact that in wet regions, where vegetation is less constrained by
water availability, changes in $S_{umax}$ are primarily influenced by other
environmental factors than $AI$ (Green et al., 2022). With climate becoming
wetter (Figure 2), i.e. when the drought index decreases, root zone storage
capacity may increase due to other factors such as rising temperature and
nutrient availability, which would lead to an increase in $NDVI$, and
consequently, greater vegetation water demand, resulting in an increase in $S_{umax}$.
This ultimately creates a negative correlation between $S_{umax}$ and $AI$.



Cluster 4, although located in a humid region, receives relatively low
precipitation, primarily due to low *AI* caused by cooler temperatures in
mountainous areas (Figure 2). Climate warming not only increased *AI* but also
enhanced vegetation productivity, jointly driving an increase in root zone water
demand. Hence, Cluster 4 tends to show a positive correlation. Only a few arid
cluster catchments (2, 8, 9) are primarily dominated by the *AI*, leading to a
positive correlation in most of the catchments.
## 4. Discussions
The comparison of the two independent approaches for estimating root zone
storage, as shown in Figure 3 shows a consistent behavior between the ERA-5
derived $S_R$ and $S_{umax}$. This result suggests that both approaches identify the same
variable, which we associate to the root zone storage capacity. There are similar
parameters determining the splitter of runoff generation and infiltration to meet
water deficit (and eventually used for evaporation during dry spells) in
hydrological models, such as the tension water capacity in the Xinanjiang model
(Zhao et al., 1992; Hu et al., 2004), the maximum soil water storage (or field
capacity in original version) in the HBV hydrological model (Lindström et al.,
1997; Seibert et al., 2022), and the maximum capacity of the production store in
the GR4J model (Perrin et al., 2003). Among these models, Xinanjiang model
used a probability distribution curve to represent the catchment characteristic of
storage capacity, thus with more solid physical foundation (Moore, 2007). That
is the reason we chose the Xinanjiang curve as root zone storage capacity
distribution in this study.
Long-term catchment-scale streamflow and spot-scale lysimeter
measurements revealed that root zone seepage matched perfectly with catchment
runoff in the Rietholzbach research catchment in Switzerland, although these
two observations have large scale discrepancy (see Figure 4 in Seneviratne et
al., 2012). Moreover, Nijzink et al. (2016) compared $S_R$ derived from water
balance with the $S_{umax}$ parameters of four hydrological models, revealing
remarkably similar patterns in the three studied catchments in the United States.
All these experimental and modeling studies using multi-source data and
independent methods further confirmed that $S_{umax}$ does represent the root zone
storage capacity.
For the trend of $S_{umax}$, we found that, over the years, $S_{umax}$ is increasing in
the United States and that this increase can be largely attributed to climate
change. This corresponds to the results of Merz et al. (2011), who used the HBV
model to simulate 273 catchments in Austria and found that the soil water
storage parameter FC nearly doubled from 150 to 275 mm in 30 years and
attributed it to increases in temperature and evaporation.
As a survival strategy, plants adopt a cost minimization in the design of
their root systems, aiming to meet the water demands of the canopy with the
minimum allocation of root carbon (Milly, 1994). In the Mediterranean climate
region with strong seasonality of precipitation, abundant rainfall during the wet
season boosts vegetation productivity, leading to a deeper rooting system (Fan et



al., 2017). During the dry season, vegetation may rely on tap roots to access groundwater (Dawson and Pate, 1996). In forest areas with sufficient water supply, rainfall thoroughly saturates the soil, and due to frequent surface wetting, the root systems do not require access to deep water. The spatial distribution of $S_{umax}$ observed in this study is consistent with the results of Gao et al. (2014a), who calculated the root zone storage capacity for over 300 catchments using Model Parameter Estimation Experiment (MOPEX) data. Both studies demonstrated the increase of $S_{umax}$ in response to an increase of the aridity index, i.e. geospatially from the humid east coast to the dry inland regions of the United States. Additionally, this study extends the analysis of $S_{umax}$ from spatial to temporal variability under changing environmental conditions.

This study compared the root zone storage capacity calculated by two different methods and datasets ($S_{umax}$ and $S_R$). Disparities were observed between the two outcomes, such as significant differences in the magnitude of trend slopes between $S_{umax}$ and $S_R$. These disparities may be attributed to the presence of croplands in certain catchments, which are heavily influenced by human activities. The MCT method accounted for these human activities, such as irrigation and artificial reservoirs, which increase water supply to the root zone during dry seasons, thereby alleviating water shortage and leading to $S_R$ reduction compared to natural ecosystems. On the other hand, the discrepancy may have resulted from scale mismatches, i.e. the $S_{umax}$ at catchment scale and the $S_R$ derived from ERA5 data (spatial resolution of 0.5 degree) used by the MCT method. It is difficult to draw solid conclusion on which method is more reliable than the other. From the perspective of methodology, both methods have a strong physical basis. But the MCT method explicitly considers the human activities, such as irrigation, on atmospheric moisture fluxes; while they have implicit impact on DYNIA results through the runoff, although it is difficult to isolate the influence of human activities. From the perspective of forcing data uncertainty, MCT method in this study is based on the ERA-5 land surface reanalysis data; while DYNIA method is based on observed hydrological data, which is normally more reliable in a catchment scale study. There may be other reasons causing the different magnitudes. We still need more studies to understand this issue and close the gap between two independent methods.

This study employed two methods to calculate root zone storage capacity, both methodologies calculated the total evaporation without differentiating between transpiration from vegetation and soil evaporation. Thus, this is a fair comparison. Moreover, soil evaporation constitutes a relatively small proportion of the terrestrial hydrological fluxes, around 6% of the total evaporation in a global scale analysis (see Good et al., 2015). This proportion is even lower in regions with vegetation cover, which is the predominant land cover in most catchments of this study. Hence, vegetation water use transpiration from root zone is an overwhelmingly major flux dominating dry spell evaporation.

Our results show a positive correlation of $S_{umax}$ with $E/P$ and a negative correlation with $Q/P$ (which in the long term equals 1- $E/P$) According to the Budyko framework (Marlatt et al., 1975; Donohue et al., 2006) (the relation between aridity $Ep/P$ and $E/P$), the division of flow into runoff and evaporation



is highly influenced by $S_{umax}$ (Cheng et al., 2017; Gentine et al., 2012; Luo et
al., 2020; Gerrits, 2009). An increase in $S_{umax}$ implies increased plant
transpiration, leading to higher $E/P$ and lower $Q/P$. As a result, catchments with
decreased $S_{umax}$ have higher $Q/P$ and *vice-versa*. By establishing dynamic
relationships between model parameters and environmental factors, it challenges
the assumption of a static modeling framework. In contrast, allowing dynamic
model parameters would allow to model the effect of environmental changes on
catchment hydrological characteristics.

## 472    5. Conclusions

In this study, we used a large sample dataset to estimate the temporal
variation of $S_{umax}$ through dynamic parameter identification of the FLEX
hydrological model. The aim was to enhance our understanding of $S_{umax}$
variation in a changing environment and to improve the model's ability to
simulate under such conditions. We found that from 1980 to 2014, $S_{umax}$ in most
catchments across the United States showed a significant increasing trend. 423
catchments (85%) showed increasing $S_{umax}$, and the average $S_{umax}$ of the 497
catchments increased from 178 to 235 mm, representing a 32% increase.

The $S_R$ obtained through the MCT method exhibited similar spatial
distribution and temporal patterns to $S_{umax}$, not only affirming the authenticity of
$S_{umax}$ growth without calibration-induced artifacts but also emphasizing that the
hydrological model parameter $S_{umax}$ indeed represents root zone storage capacity.
This indicates that the constantly changing climate significantly changes the
ecohydrological processes of the catchment, compelling vegetation to adjust its
root zone storage capacity to adapt to the environment.

Furthermore, the temporal correlation analysis between $S_{umax}$ and
environmental factors reveals a significant negative correlation between $S_{umax}$
and both runoff and runoff coefficient. This indicates a strong connection
between ecosystem dynamics and hydrological processes. In summary, using
multi-source datasets and independent methods, we found a significant increase
of root zone storage capacity in the United States, indicating ecosystems'
adaptation of belowground biomass in response to environmental change. It
shows that it is important to consider a dynamic root zone in hydrological and
land surface modeling studies.

**Competing interests**
At least one of the (co-)authors is a member of the editorial board of
Hydrology and Earth System Sciences.

**Acknowledgements**
This research has been supported by the National Natural Science
Foundation of China (grant no. 42071081 and 42122002).



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




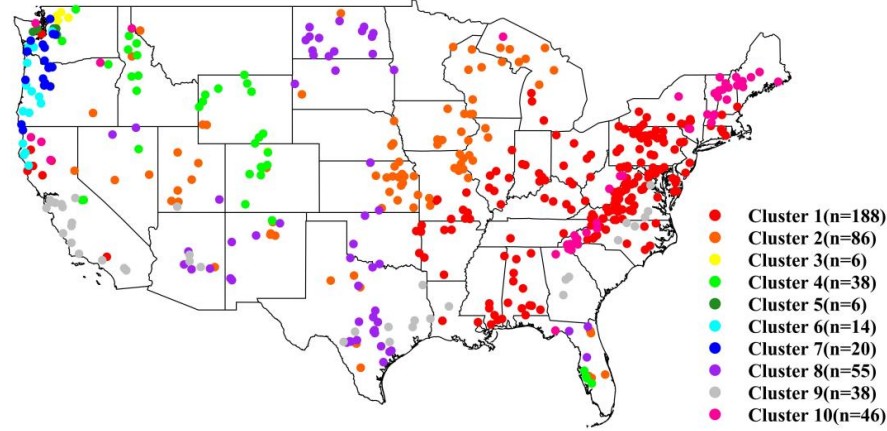


**Figure 1** Maps of the 497 CAMELS catchments in the United States, adopted the
clusters provided by Jehn et al. (2020).

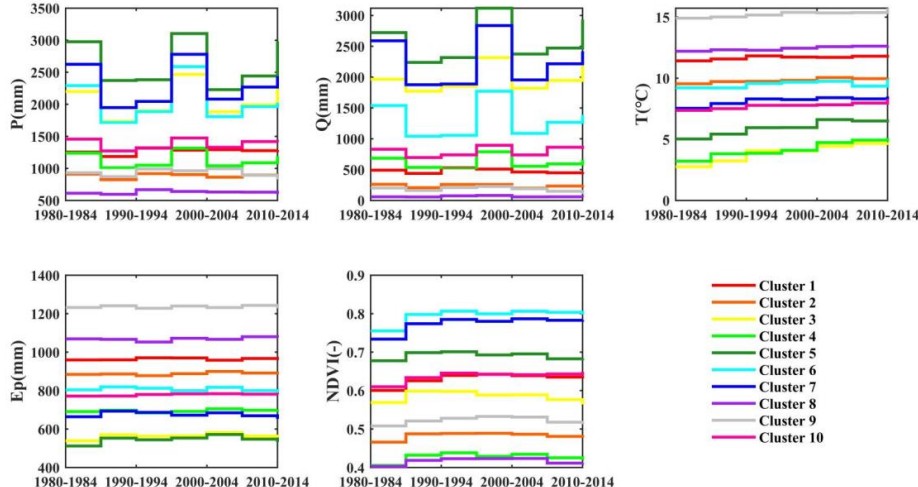


**Figure 2** Five-year average temporal trends (mean values) for 10 clusters of
precipitation $P$, runoff $Q$, temperature $T$, potential evaporation $Ep$, and $NDVI$.

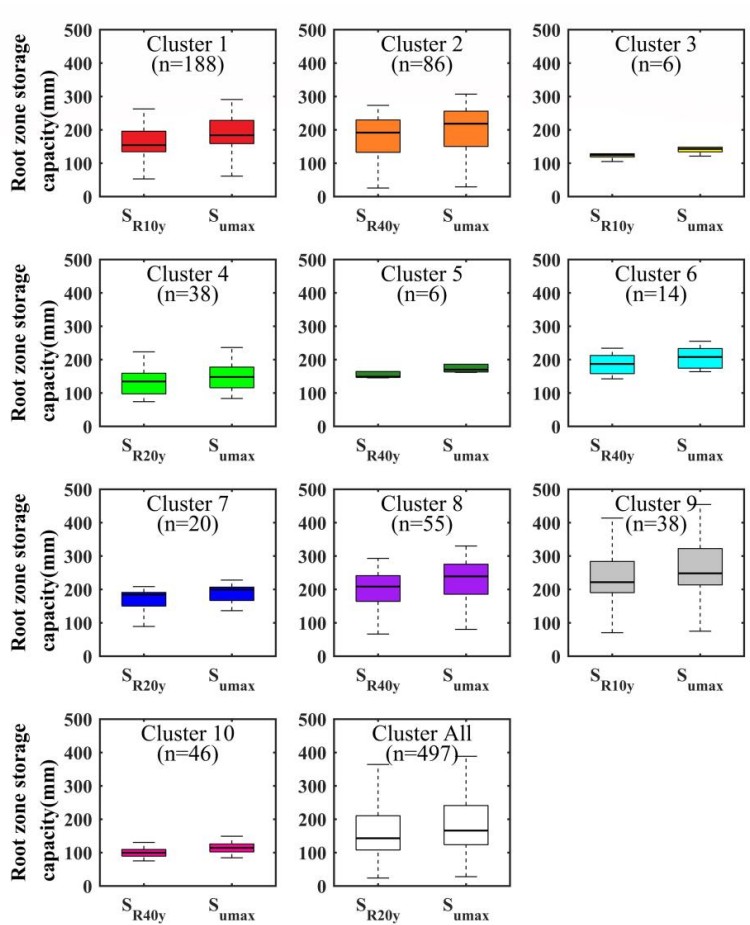


**Figure 3** The Box-Whisker plots display the spatial distribution of $S_{umax}$ and $S_R$
(including $S_{R10y}$, $S_{R20y}$, and $S_{R40y}$, representing the required root zone storage capacity
to overcome certain return periods of droughts, i.e. 10, 20 and 40 years) across 497
study catchments within 10 clusters. The bottom and top edges of the box representing
the 25th and 75th percentiles, respectively. The solid lines represent the median
values, while the upper and lower whiskers extend to the furthest data points that are
not outliers.



**Figure 4** The temporal variations of 10 parameters across 497 study catchments within 10 clusters. The blue solid line represents the average change trend of parameters for all catchments within each cluster. The black dashed line indicates the fitted regression line. The gray shaded area represents the 20%–80% envelop. Standardized regression coefficients are marked in red (above each panel). The significance of regression coefficients is indicated as: '' for 0.1, '' for 0.01, and '' for 0.001.



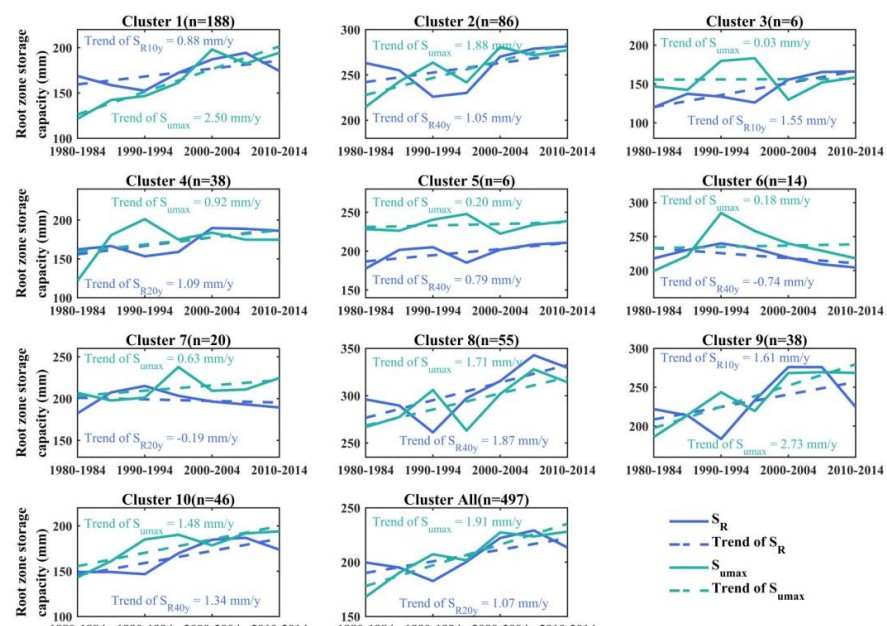

**Figure 5** The temporal variations of $S_{umax}$ and $S_R$ (including $S_{R10y}$, $S_{R20y}$, and $S_{R40y}$, representing the required root zone storage capacity to overcome certain return periods of droughts, i.e. 10, 20 and 40 years) across 497 study catchments within 10 clusters. Solid lines represent the average change trend of $S_{umax}$ or $S_R$ for all catchments within each cluster. Dashed lines indicate the fitted regression lines, with corresponding regression coefficients marked in the same color.



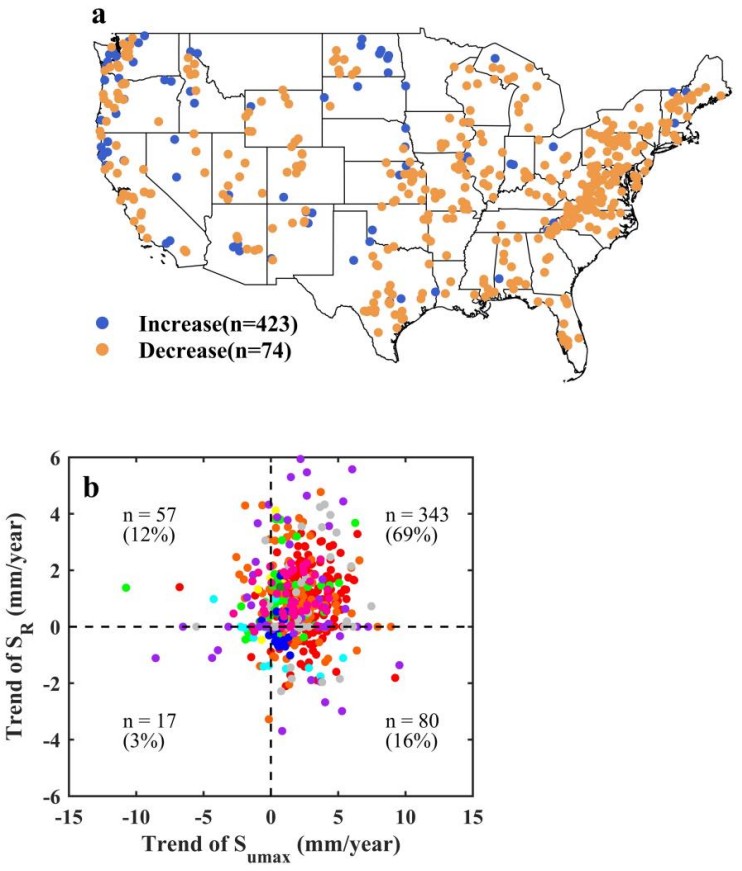

**Figure 6** (a) The trend of $S_{umax}$ variation across 497 study catchments: 423 (85%) catchments exhibit an increasing trend, while 74 (15%) catchments show a decreasing trend. (b) The comparative trends of $S_{umax}$ and $S_R$ across 497 study catchments. Different colored points represent clusters.





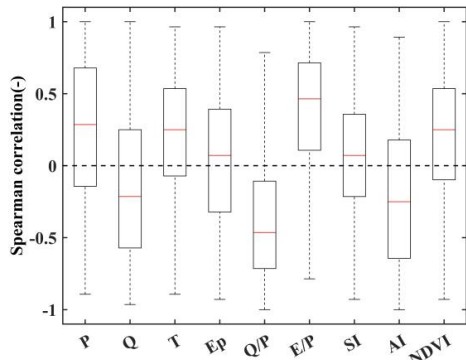

**Figure 7** The Box-Whisker plot displays the Spearman temporal correlation coefficients between the $S_{umax}$ model parameters and environmental elements for 497 catchments over a calibration period of seven 5-year cycles. Precipitation $P$, runoff $Q$, temperature $T$, potential evaporation $Ep$, runoff coefficient $Q/P$, evaporation coefficient $E/P=1-Q/P$ (assuming the delta of water storage at annual scale is small), aridity index $AI$, precipitation seasonality index $SI$. The bottom and top edges of the box representing the 25th and 75th percentiles, respectively. The solid red lines represent the median values, while the upper and lower whiskers extend to the furthest data points that are not outliers.





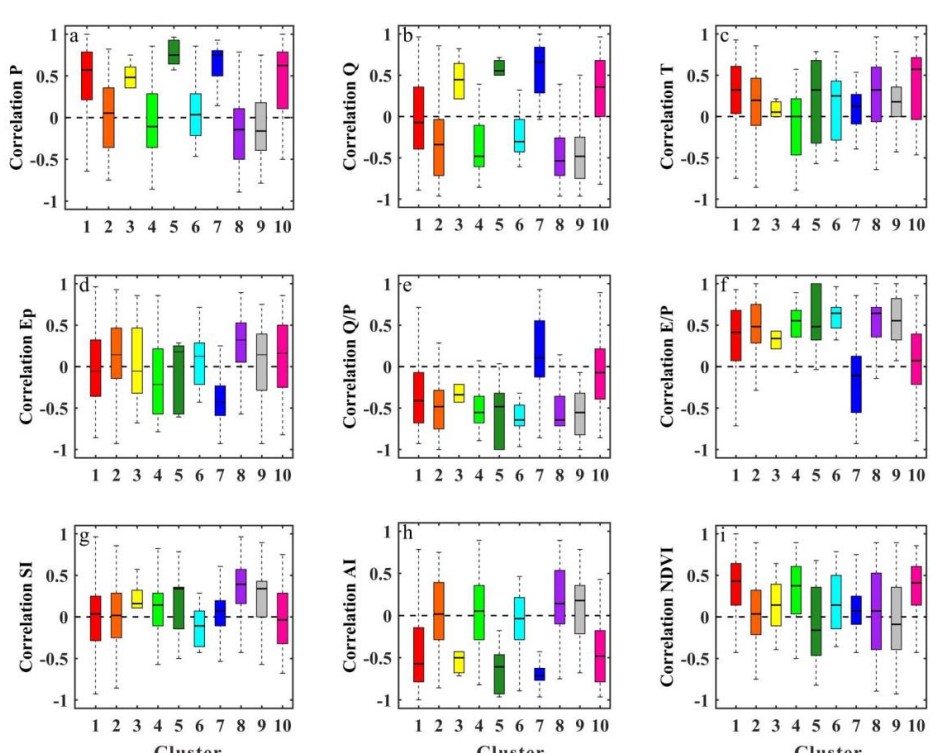

**Figure 8** The Box Whisker plot displays the Spearman temporal correlation coefficients between $S_{umax}$ and environmental elements over a calibration period of seven 5-year cycles across 10 clusters of 497 research catchments. The bottom and top edges of the box representing the 25th and 75th percentiles, respectively. The solid lines represent the median values, while the upper and lower whiskers extend to the furthest data points that are not outliers.



**Table 1** Properties of catchment clusters (Jehn et al., 2020).

| Cluster | Number of catchments | Main region | Dominating attribute |
|---|---|---|---|
| 1 | 188 | Southeastern and Central Plains | Aridity |
| 2 | 86 | Central Plains (with scattered catchments all over western US) | Green vegetation fraction maximum |
| 3 | 6 | Northwestern Forested Mountains | Fraction of precipitation falling as snow |
| 4 | 38 | Northwestern Forested Mountains and Florida | Precipitation seasonality |
| 5 | 6 | Northern Marine West Coast Forests | Forest fraction |
| 6 | 14 | Marine West Coast Forests | Aridity |
| 7 | 20 | Western Cordillera (Part of Marine West Coast Forests) | Fraction of precipitation falling as snow |
| 8 | 55 | Great Plains and North American deserts | Precipitation seasonality |
| 9 | 38 | All southernmost states of the US | Aridity |
| 10 | 46 | Appalachian Mountains | Mean elevation |



**Table 2** FLEX model water balance equations and structural equations.

| Sub-module | Water balance equations | Constructive equations |
|---|---|---|
| Snow reservoir | $\dfrac{dS_W}{dt} = \begin{cases} -M, & T > T_t \\ P_s, & T \le T_t \end{cases}$ (1) | $M = \begin{cases} \min(S_w, F_{DD}(T - T_t)), & T > T_t \\ 0, & T \le T_t \end{cases}$ (2) |
| Interception reservoir | $\dfrac{dS_i}{dt} = P_r - E_i - P_{tf}$ (3) | $E_i = \begin{cases} E_p, & S_i > 0 \\ 0, & S_i = 0 \end{cases}$ (4) $\qquad$ $P_{tf} = \begin{cases} 0, & S_i < I_{max} \\ P_r, & S_i = I_{max} \end{cases}$ (5) |
| Unsaturated soil reservoir | $\dfrac{dS_u}{dt} = P_e - E_a - R_u$ (6) | $E_a = (E_p - E_i)\,min\left(\left(\dfrac{S_u}{S_{umax}C_e}\right), 1\right)$ (7) $$R_u = \begin{cases} P_e - S_{umax} + S_u + S_{umax}\left(1 - \dfrac{P_e+AU}{(1+\beta)S_{umax}}\right)^{(1+\beta)}; & (1+\beta)S_{umax} > P_e + AU \\ P_e - S_{umax} + S_u; & (1+\beta)S_{umax} \le P_e + AU \end{cases}$$ (8) $$AU = (1+\beta)S_{umax}\left(1 - \left(1 - \dfrac{S_U}{S_{umax}}\right)^{\left(\frac{1}{1+\beta}\right)}\right)$$ (9) |



| Splitter and lag function | $R_f = R_u D (10); R_s = R_u (1 - D) (11)$ |
| | $R_{lf} = \sum_{i=1}^{T_{lag}} c(i) \cdot R_f(t - i + 1) \ (12)$ |
| | $c(i) = i = \sum_{i=1}^{T_{lag}} u \ (13)$ |
| Fast reacting reservoir | $\dfrac{dS_f}{dt} = R_{fl} - Q_f (14)$ $\qquad Q_f = S_f / K_f (15)$ |
| Slow reacting reservoir | $\dfrac{dS_s}{dt} = R_s - Q_s (16)$ $\qquad Q_s = S_s / K_s (17)$ |



**Table 3** Description of FLEX model parameters and range of values.

| Parameter | Explanation | Range | Units |
|-----------|-------------|-------|-------|
| $F_{DD}$ | Degree day factor | 1-7 | mm/(d°C$^{-1}$) |
| $T_t$ | Threshold temperature | -2-4 | °C |
| $I_{max}$ | Maximum $S_i$ storage | 1-5 | mm |
| $S_{umax}$ | Root zone storage capacity | 30-700 | mm |
| $C_e$ | Threshold of soil moisture content | 0.1-1 | - |
| $\beta$ | Spatial diversity factor | 0-1 | - |
| $D$ | Splitter factor | 0-1 | - |
| $K_f$ | Fast runoff timescales | 1-10 | d |
| $K_s$ | Slow runoff timescales | 10-200 | d |
| $T_{lag}$ | Lag-time between storm and fast runoff | 0.8-10 | d |