# Peer review of "Widespread increase of root zone storage capacity in the United States"

_EGUsphere, 2024_

## Author Comment (AC1)

Summary and general comments

Overall, the paper is very well structured. The language is good but sometimes a bit imprecise, so the paper would benefit from a thorough reread. The topic is of interest to the HESS readership and has the potential to become a very interesting contribution, but it requires major revisions before it can be published.

As a side note, I would always encourage authors to include tables and figures in the text. This makes it easier to review.

Reply: We thank Anonymous Referee #1 for his/her careful review and valuable feedback. We appreciate his/her endorsement of the paper's "potential to become a very interesting contribution". This affirmation has greatly encouraged us, and the provided suggestions have provided clear guidance for improvement.

We take the reviewer concern regarding the precision of the language seriously. We will undertake careful revisions to ensure clearer and unambiguous expression. We will insert figures and tables at appropriate positions in the main text for easier reading rather than at the end.

Major comments

While the two methods use different datasets and approaches, I wonder how independent they really are. If the forcing data roughly agree (especially on longer than daily timescales and trends), then they contain similar information. The core of the FLEX model is more complex than a simple water balance, but follows a similar idea. The phrasing "multi-sources data and independent methods" made me expect a bit more than "just" another water balance approach, perhaps some independent observations/estimates of vegetation dynamics or rooting depths. I would thus suggest to rephrase "validated by multi-sources data and independent methods" in the abstract and the corresponding parts in the manuscript and state right away that these two methods were used. And I would encourage at least a discussion on other kinds of more independent data that could be used to evaluate the findings.

Reply: We will clarify what we mean by "multi-sources data and independent methods". Indeed, it was insufficiently clear from the original submission that the two methods use different data. In particular, the $S\_umax$ calibration approach within FLEX using DYNIA is constrained by streamflow data. The MCT method, instead, does not use streamflow data, but only meteorological variables (P and E). Moreover, the approach using FLEX makes use of the Daymet dataset. The MCT method uses ERA-5 reanalysis data. We will follow the recommendation to clarify these differences early on in the paper such as in the abstract and introduction.

In the manuscript, it sometimes sounds like it is obvious that this parameter (and the MCT estimate) actually relates to root zone storage capacity, but somehow I'm still a little unsure (or perhaps unconvinced) about the physical meaning of the parameter. So, I think a bit more

discussion on the physical meaning of these estimates would be good. For instance, in l.69 it says "such problems can be attenuated through specific modelling choices…" but later the problem of parameter identification and the difficulty of relating parameters to measurable properties (i.e. the root zone) isn't discussed much.

Related to that, I was wondering if the authors ever tried to use the MCT on model outputs. It would be interesting to see if one could infer the then known parameter $S_{umax}$ from an idealized modelling situation using the MCT.

Reply: We will clarify that $S_{umax}$ is a conceptual parameter, in both approaches (FLEX and MCT). We speculate that this parameter has also a physical meaning, expressing a fundamental characteristic of integrated catchment behavior. In particular, it represents the maximum storage available to vegetation water uptake. We will make clear that this is an interpretation rather than a result of our study.

Regarding the question of using MCT estimates in the hydrological model, there have several researches have used $S_R$ estimated by MCT to simulate runoff which documented that the accurate estimation of $S_R$ can improve hydrological and land-surface model performance (Bouaziz et al., 2022; Sakschewski e al., 2021; Mao & Liu, 2019; Wang et al., 2021).

We did try to replace the model's $S_{umax}$ parameter with the output from MCT. However, the model simulation results were not as expected. We think there could be at least two reasons: 1) the model's $S_{umax}$ parameter was obtained by calibration with optimization algorithm based on streamflow data. And $S_R$ by MCT was derived from meteorological data only. Hence, the inferior performance of $S_R$ is actually not a surprise. 2) the catchment boundary does not perfectly match with the ERA-5 reanalysis data, due to the relatively small catchment areas and the coarse resolution of ERA-5 data with 0.5 degree. So, we ultimately did not include this part in the paper. But we will add some discussion in the revised manuscript reflecting this attempt.

In the future research, we will consider to use higher resolution reanalysis data or remote sensing data to estimate $S_R$ using MCT and use it in model to simulate runoff.

Reference

L. J. E. Bouaziz, E. E. Aalbers, A. H. Weerts, M. Hegnauer, H. Buiteveld, R. Lammersen, J. Stam, E. Sprokkereef, H. H. G. Savenije, M. Hrachowitz, Ecosystem adaptation to climate change: the sensitivity of hydrological predictions to time-dynamic model parameters. Hydrol. Earth Syst. Sci. 26, 1295-1318 (2022).

B. Sakschewski, W. von Bloh, M. Drueke, A. A. Soerensson, R. Ruscica, F. Langerwisch, M. Billing, S. Bereswill, M. Hirota, R. S. Oliveira, J. Heinke, K. Thonicke, Variable tree rooting strategies are key for modelling the distribution, productivity and evapotranspiration of tropical evergreen forests. Biogeosciences 18, 4091-4116 (2021).

G. Mao, J. Liu, WAYS v1: a hydrological model for root zone water storage simulation on a 20 global scale. Geosci. Model Dev. 12, 5267-5289 (2019).

Wang, J., Gao, H., Liu, M., Ding, Y., Wang, Y., Zhao, F., and Xia, J.: Parameter regionalization of the FLEX-Global hydrological model, Sci. China Earth Sci., 64, 571-588, https://doi.org/10.1007/s11430-020-9706-3, 2021.

I am a bit surprised by the results. If I am not mistaken, S_umax increases with aridity (PET/P) in Gao et al. (2014), but the temporal trend here shows the opposite. Why is that? Why would an increase in P lead to larger S_umax? Wouldn't more precipitation lead to a smaller root zone? Does a concurrent increase in P and PET (an intensification if you will), require more storage to maximize ET? Or is the increase in P caused by more intense precipitation events that require more buffering potential in the root zone? Related to the last question, I was wondering why storm characteristics (e.g. inter-storm duration, event intensity) were not analyzed. Inter-storm duration was a strongly correlated climate attribute in Gao et al. (2014), so it would be interesting to see how that correlates with the observed changes, also given that it might change in the future. Overall, I think the discussion about spatial and temporal relationships between root zone storage capacity and climate attributes requires more depth and also a clearer description of the hypothesized processes. I think currently not all statements are totally coherent and the some of the statements are a bit vague or only loosely relate to the findings of the paper.

Reply: Excellent point. Regarding the relationship between S_umax and aridity (Ep/P), we provided an explanation in lines 355-379 of the paper, indicating that S_umax decreases with increasing aridity over time. This may be due to other factors besides aridity dominating the temporal variation of root zone storage capacity in these basins. He/she point about increased precipitation leading to a decrease in S_umax is valid, while the results of this study show the opposite. According to our analysis, approximately two-thirds of the basins exhibited an increase in evaporation rate greater than the growth rate of precipitation, indicating an increase in the evaporative coefficient. This resulted in the positive impact of evaporation on S_umax growth exceeding that of precipitation. Hence, contrary to expectations, there appears to be a positive correlation between precipitation and S_umax. We did not address this in the manuscript, and we appreciate he/she highlighting it. We will incorporate this discussion into the subsequent revisions of the paper. Regarding the correlation analysis between inter-storm duration and S_umax, inter-storm duration only considers days with zero precipitation. However, on days with only weak precipitation, water may evaporate before reaching the root zone. Therefore, it is more appropriate to consider the number of days where evaporation exceeds precipitation. Since we lack actual evaporation data, we conducted tests using potential evaporation instead. Specifically, we calculated the maximum number of days where potential evaporation exceeded precipitation, which indeed showed a positive correlation with S_umax. However, as this data does not represent actual evaporation, we did not include this result in the article.

Generally, I think the discussion would benefit from some major edits. Some of the paragraphs do not relate much to the findings (e.g., l.396-405 or l.451-459 do not add much for me) and could be shortened or removed. Instead, I would find it interesting to see more discussion on the physical mechanisms that drive a change in root zone storage capacity (see above). Why would an increase in P lead to an increase in S_umax? Other questions that would be interesting are: How could we actually set the S_umax parameter dynamically if we do not have calibration data? What other independent data sources could be used to evaluate the findings of the study?

Reply: Thank you very much for his/her valuable feedback. We think that the content in l.396-405 elucidates that S_umax represents the root zone storage capacity, which is crucial for a deeper understanding of the physical meaning of S_umax. However, we recognize that the current expression may not be clear enough, so we plan to revise this section to more explicitly explain the relationship between S_umax and root zone storage capacity. The l.451 to 459 explains that both methods used to calculate root zone storage capacity did not differentiate between vegetation transpiration and soil evaporation in total evaporation. Given that this point is crucial for understanding the effectiveness of the methods, we believe that this information may be more suitable to be included in section 2 Data and Methods. Therefore, in the subsequent revision, we plan to move this discussion to section 2. The reason for the positive correlation between precipitation and S_umax has been explained in the previous question. We agree with his/her suggestion to include more discussion on the spatial and temporal relationships between S_umax and catchment characteristics. At present, six types of approaches to estimate the root zone storage capacity have been suggested or are in use in hydrological and land surface models: the field observation-based approach, the look-up table approach, the optimization approach, the inverse modelling approach, the calibration approach, and the mass balance based approach (Wang- Erlandsson et al., 2016). We think that lysimeter data is an independent data source to evaluate the study's findings, but we still need to confirm the availability of such data.

Minor comments

Abstract and also l.39: "The root zone is the upper part of the unsaturated zone···" – What if roots are below the water table?

Reply: He/she is correct in pointing out that roots do exist below the water table, but they are relatively scarce (Fan et al., 2017). Typically, roots develop above the water table, within the unsaturated zone. In the previous abstract and main text, we may have oversimplified the description of root location without fully considering this specific scenario. We will consider refining the description in the revision to more accurately portray this phenomenon.

Reference

Fan, Y., Miguez-Macho, G., Jobbágy, E. G., Jackson, R. B., and Otero-Casal, C.: Hydrologic regulation of plant rooting depth, Proc. Natl. Acad. Sci. U. S. A., 114, 10572–10577, https://doi.org/10.1073/pnas.1712381114, 2017.

Abstract: "controlling the partitioning into storage, runoff and percolation" and *transpiration*? This seems to be the most important flux that depends on the root zone.

Reply: Partitioning and transpiration are two processes occurring in two different time scale.

Partitioning happens in rainfall event. Rain water increases root zone storage, which is used for transpiration during dry spells. Thus partitioning is a very short time period, while transpiration lasts for a much longer dry spells periods. Hence, we would keep the text as it is.

Abstract: In the abstract, it sounds like the main focus is on the model-derived S_umax, but in the main text it sounds like this approach and the MCT approach are both equally important for the analysis. I would suggest reporting the numbers in l.330-341 also in the abstract and to add to the abstract that the MCT was used as a second method, rather than just writing "validated by multi-sources data and independent methods".

Reply: Agreed with his/her suggestion. Indeed, in our study, both the MCT method and the model-derived S_umax play important roles, forming the foundation of our analysis. We will add the MCT results from l.330-341 to the abstract and conclusion in subsequent revisions.

l.39: "in which vegetation buffers water" – I am not sure if I'd say that vegetation buffers water, because it's stored in the soil/weathered rock after all

Reply: We agree. Our intention was to emphasize the important role of the root zone in storing water, but the expression was not accurate enough. Thank you for pointing that out. We will revise 'in which vegetation buffers water' to 'in which root zone buffers water'.

l.41: "the real shape" – what do the authors mean here? The shape of the bucket? Or just the value of S_umax?

Reply: Thank you for pointing out that "the real shape" refers to the actual shape of "the root zone bucket". The expression in the text may not be clear, and we will revise this part accordingly.

l.57: It is perhaps costly but one could think of measuring rooting depths etc. over time (using many similar plots etc.).

Reply: Thank you for his/her suggestion. We mean that the method based on soil texture cannot reflect the dynamic response of the S_R to climate change because of rare variation in soil texture during short term. The deficiencies of using rooting depth contain not only the labor cost but also the incomplete proxy for S_R. Although these two terms have strong connection, they are definitely two different things as in detail discussed by Gao et al. (2024). While rooting depth only describes the vertical extension of roots, it does not consider root distribution along the profile. Root zone, furthermore, accounts for lateral root extent and root density. For example, an ecosystem covered by deep-rooting vegetation with roots with low spatial density likely has a smaller root zone than one covered by vegetation with shallow roots with higher spatial densities (Singh et al., 2020; van Oorschot et al., 2021).

References

Gao, H., Hrachowitz, M., Wang-Erlandsson, L., Fenicia, F., Xi, Q., Xia, J., Shao, W., Sun, G., and Savenije, H. (2024). Root zone in the Earth system, EGUsphere [preprint], https://doi.org/10.5194/egusphere-2024-332.

Hauser, E., Sullivan, P. L., Flores, A. N., Hirmas, D. & Billings, S. A. (2022). Global-scale shifts in rooting depths due to Anthropocene land cover changes pose unexamined consequences for critical zone functioning. Earth Future, 10, e2022EF002897.

Singh, C., Wang-Erlandsson, L., Fetzer, I., Rockström, J. & van der Ent, R. (2020). Rootzone storage capacity reveals drought coping strategies along rainforest-savanna transitions. Environmental Research Letters, 15, 124021.

van Oorschot, F., van der Ent, R. J., Hrachowitz, M., and & Alessandri, A. (2021). Climate-controlled root zone parameters show potential to improve water flux simulations by land surface models. Earth System Dynamics, 12, 725-743.

l.76: Hauser et al. (2022) estimated global changes in rooting depths. The study might be relevant here.

Reply: The study by Hauser et al. (2022) provided a good understand about the adaption of rooting depth to climate and human activities. However, $S_R$ is distinct from rooting depth. $S_R$ shows the hydrological adaption to climate, especially the drought here, using a water amount to represent information more than rooting depths, such as root density and lateral extent. That will help better understand ecosystem hydroclimate adaption. We will emphasize the hydrological consideration by using expression "little is known about terrestrial ecosystems' root zone hydrology adaption to these changes''.

l.94: Perhaps also add Newman et al. (2015), which was a precursor to CAMELS US.

Reply: He/she is correct, and we will cite this paper in the revised version.

l.101: Was PET calculated by the authors or taken from CAMELS? I also assume that NDVI is not from CAMELS? Please clarify.

Reply: The potential evaporation was calculated using temperature data from the CAMELS dataset based on Hargreaves equation (Hargreaves et al., 1985) method. We will revise it accordingly. NDVI data is not sourced from the CAMELS dataset, it comes from NOAA GIMMS. Details about its specific source can be found in l.104.

Figures 1,2: Finding colors for so many variables is difficult, but there should be better color palettes than this one with all the greens and reds.

Reply: Thank you for his/her valuable feedback. Given the large number of clusters, ensuring diversity and distinctiveness in color selection is indeed a challenge. Here, we referred to the color schemes proposed by Jehn et al., 2020, and Mathai & Mujumdar, 2022. If he/she have specific color suggestions or preferences, please feel free to share them with us.

Figure 2: I was not entirely clear to me over which time windows the variables were averaged. And what are the units? mm/yr?

Reply: In terms of time windows, the variables are averaged over the following periods: 1980-1984, 1985-1989, 1990-1994 (on the x-axis), and so forth. As for the units of the variables, it depends on the data type presented in Figure 2, which is indicated on the y-axis.

Table 2 and 3 and the model description could be moved to an appendix. Instead, one summary paragraph could be added to the methods that explains the most important aspects of the models, incl. relevant parameters. For instance, I do not see why the routing parameters are explained, as they shouldn't affect the partitioning in the root zone.

Reply: Thank you for his/her suggestion. Regarding the suggestion to move Tables 2 and 3 along with the model description to an appendix, we understand this is to simplify the main text and highlight the core content. However, considering the importance of these tables and descriptions for a comprehensive understanding of the research methods and model details, we believe keeping them in the main text will help readers delve deeper into our study. As for the routing parameters he/she mentioned, while they may not directly impact the partitioning in the root zone, they play a crucial role in model construction and simulation processes. For the completeness and accuracy of the study, we think it is necessary to explain them.

l.179: Was the model performance assessed over 5 years and was the window then also moved by 5 years or only by 1 year? That was not completely clear to me.

Reply: Thank you for pointing this out. In the process of model performance evaluation, we actually moved in 5-year time windows instead of 1 year. We shall express this point clearly enough in the text.

l.198: How exactly did the authors define net precipitation? Rain (i.e. liquid precipitation) minus interception?

Reply: Precipitation includes rainfall and snowfall. The term "net precipitation" here just represents rainfall minus interception. Snowfall is considered by snowmelt (SM) in MCT. We will express it more clearly.

l.215: Please check the sentence "continues in the end of the end".

Reply: This is a typo. The correct expression should be "continues at the end of the year".

Methods: Was the MCT curve applied to the same catchments by extracting the fluxes from ERA5 via catchment shapes? This was not clear to me.

Reply: No. In this version, we calculated the global S_R values using ERA5 grid-scale data. Then, based on the catchment shapes provided by the CAMELS data, we further extracted

the corresponding S_R values for each catchment. However, his/her suggestion reminds us that we should first extract fluxes from ERA5 data through catchment shapes and then calculate catchment S_R to more accurately reflect the hydrological characteristics within the catchment. We will recompute S_R based on this suggestion.

l.219-231: The selection of the 10yr, 20yr, 40yr, etc. based on vegetation type seems a bit arbitrary. It would be good to add some references and possibly also an uncertainty analysis.

Reply: We referred to the studies of Gao et al. (2014) and Wang-Erlandsson et al. (2016) to select the return periods. According to these studies, we simply selected the 5, 10, 20, 30 and 40 years as return periods. In fact, it is also a meaningful work for hydrology to determine the suitable return periods by comparing S_R to S_umax. Thank you for his/her suggestion. We will add explanations and references regarding the selection criteria for these return periods.

l.227: How exactly did the authors calculate S_r for different return periods? Maybe add a few more explaining sentences. Are the return periods chosen (e.g. in Figure 3) just based on the vegetation type? Or are the return periods in Figure 3 chosen based on the results, so that they fit best? If this is too much repetition from Gao et al. (2014) it could be added to an appendix.

Reply: The method for selecting return periods in Figure 3 is similar to that used by Gao et al. (2014), both being based on the minimum difference between S_R and S_umax. While Gao et al. (2014) utilized the MOPEX dataset and primarily focused on discussing the spatial distribution of root zone storage capacity, our study builds upon their work by not only examining the spatial distribution but also emphasizing the analysis of its temporal variations.

l.237: If the evaporation coefficient is calculated using Q/P, doesn't it contain exactly the same information as the runoff coefficient? The correlations etc. seem to be exactly the same, so I don't see why this is added. And why 7 data points per regression (which really is not a lot to calculate a trend).

Reply: The root zone is an important layer for water storage and exchange, determining how precipitation is distributed within a catchment as runoff and evaporation. The evaporation ratio (E/P) is a key indicator for assessing the proportion of precipitation converted into evaporation; a higher value suggests a greater proportion of precipitation returning to the atmosphere through evaporation. Meanwhile, the runoff ratio (Q/P) reflects the proportion of precipitation transformed into surface runoff; an increase in its value indicates more precipitation leaving the catchment in the form of runoff. These two ratios carry different physical meanings and are both related to the root zone storage capacity. Therefore, we choose to retain both of these ratios. The fact that each regression only has seven data points is related to the length of our data. Our data covers the time range from 1980 to 2014, and we use a five-year time window, resulting in exactly seven data points.

l.239: Please define aridity and seasonality index.

Reply: Thank you for his/her feedback. The aridity is represented by Ep/P and the seasonality index is represented by $SI = \frac{1}{P_a} \sum_{m=1}^{m=12} | \overline{P_m} - \frac{\overline{P_a}}{12} |$, where $\overline{p_a}$ is the mean precipitation of month m, and $\overline{P_a}$ is the mean annual precipitation. In subsequent revisions, we will ensure clear definitions.

l.248: I'd suggest deleting "unique".

Reply: Agree. We will remove the term "unique" in the subsequent revisions.

l.250: Do the authors mean "cluster averages" when they say "spatially mean"?

Reply: The terms "cluster averages" is indeed better. To avoid confusion and misunderstanding for readers, we will unify the terms in the revision.

Figure 3 and l.269: This figure does not really show the spatial distribution, which to me sounds like a map, but rather the different clusters. I word suggest changing the phrasing here.

Reply: Thank you for pointing that out. Actually, Figure 3 displays boxplots of the distribution of different cluster S_umax and S_R values. We will remove 'spatial' in the version.

Clusters: some clusters are really small (n=6) while others are large (n=188). For example, the seven smallest clusters have fewer catchment than the largest. Does that affect the results and if yes, how?

Reply: The number of catchments varies significantly among different clusters, which may influence the results. However, it is still reliable when all catchments in a small cluster showed consistent result.

l.329: The authors suddenly talk about a widespread increase, but so far no temporal changes have been presented.

Reply: We believe that, according to the content of the article, the observation that S_umax shows an increasing trend in 85% of the catchments has already been mentioned earlier in this paragraph (l. 325). We will add clear time range in the sentence.

Are the dynamics of S_umax only modelled with FLEX or also with the MCT? How was S_r calculated for 40yr return periods using a 5 year window?

Reply: No, S_umax is only modeled with FLEX. S_r is calculated using the MCT method. For calculating the S_r for a 40yr return periods using a 5-year window, we first computed the S_r value for each year within the period of 1980 to 2014 for a 40yr return periods using Gumbel

distribution. Then, every five years, we averaged these $S_r$ values to obtain the average $S_r$ value based on a 5-year window. For example, if we compute the $S_{R40}$ for each year from 1980 to 1984 and then calculate the average of these values, we obtain the average $S_{R40}$ for the period from 1980 to 1984. However, we found this method flawed and plan to improve it in subsequent versions. We will first calculate the maximum annual water deficit for each year and then, using a five-year window, compute the $S_r$ for different return periods of drought, such as a 40yr return period drought, using the Gumbel distribution. We will add explanation in methods (2.3).

Figure 4: I'd suggest to just show the important parameters here. Or rather move this figure to the appendix (like the model description).

Reply: Thank you for his/her suggestion. We may politely choose to retain Figure 4 in the main text for two reasons. Firstly, to maintain the integrity of the article, Figure 4 visually demonstrates the changes in all parameters of the FLEX model. Secondly, as the variation of $S_{umax}$ in some catchments may be influenced by other parameters, Figure 4 contains detailed information that aids in understanding these impacts.

l.368: drought index = aridity? Also, "humidity" is used in l.361. If this refers to the aridity index, I would suggest to always use aridity to have consistent terminology.

Reply: Thank you for his/her comment. We will use "aridity index" to ensure consistency in terminology in the subsequent revisions.

l.385: "splitter" seems like an odd wording here.

Reply: In subsequent revisions, we will replace the word 'splitter' as "There are similar parameters determining runoff generation or infiltration to meet water deficit……".

l.396-405: I am not quite sure what the take home message of this paragraph is.

Reply: Our intention was to emphasize that the model parameter $S_{umax}$ represents the storage capacity of the catchment's root zone. However, it appears that due to lack of clarity or specificity in our expression, this key information might not have been effectively communicated to the readers. To ensure clarity and comprehensibility, we will revise this accordingly.

l.424: "Both studies demonstrated the increase of $S_{umax}$ in response to an increase of the aridity index. But Figure 7 and the associated results section states the opposite, doesn't it?

Reply: He/she is correct. Theoretically, the availability of vegetation water is influenced by the humidity of the catchment, with larger $S_{umax}$ observed in regions of higher aridity. However, the trend of $S_{umax}$ was negatively correlated with the AI in most of the catchments in the clusters (1, 3, 5, 6, 7, 10) in wetter regions. This may be explained by the fact that in wet

regions, where vegetation is less constrained by water availability, changes in S_umax are primarily influenced by other environmental factors than AI. With climate becoming wetter, i.e. when the drought index decreases, root zone storage capacity may increase due to other factors such as rising temperature and nutrient availability, which would lead to an increase in NDVI, and consequently, greater vegetation water demand, resulting in an increase in S_umax. This ultimately creates a negative correlation between S_umax and AI.

l.434: What do the authors mean by human activities? There shouldn't be much irrigation in the CAMELS catchments.

Reply: Based on the information he/she provided, we do need to revise the description of 'human activities' in l.434.

l.461: "According to the Budyko framework ⋯ is highly influenced by S_umax". That is not true, the Budyko framework (at least Budyko's original ideas) states that it is only aridity that controls the partitioning into streamflow and evaporation, and there is no storage parameter involved.

Reply: Indeed, while the Budyko framework primarily emphasizes the impact of aridity on runoff and evaporation, storage capacity may potentially influence the Budyko curve (Zhang et al., 2001). Under same aridity conditions, an increase in storage capacity can result in a decrease in runoff and an increase in evaporation. Therefore, differences in storage capacity could lead to fluctuations in the Budyko curve.

Reference
Zhang, L., Dawes, W.R., & Walker, G.R. (2001). Response of mean annual evapotranspiration to vegetation changes at catchment scale. Water Resources Research, 37, 701 - 708. https://doi.org/10.1029/2000WR900325

l.470: How would the authors suggest to use dynamic parameters if we do not have calibration data?

Reply: This is a great question. There are at least two approaches we can try and test. 1) The empirical relationship between root zone storage capacity and climate factors can be used to predict S_umax dynamic based on future climate data (Gao et al., 2014); 2) The space-for-time assumption. Singh et al (2020) found several rainforest-savanna transitions, indicating root zone storage capacity drought coping strategies. Using this spatial pattern, we can infer S_umax temporal dynamic in future climate change. We believe this is an important scientific question worthwhile to further investigating for future studies. We will add this in the discussion.

Discussion: it might be interesting to connect the results a bit more to literature that does not come from the hydrology community and provides a somewhat complementary view on the topic, e.g. Tumber-Dávila et al. (2022).

Reply: Thank you for sharing the article. We will do more comprehensive discussion on this topic.

On the term runoff: I would suggest using streamflow if you talk about what is measured at a gauge, because it is less ambiguous.

Reply: Agreed. In subsequent revisions, we will follow this suggestion to replace 'runoff' with 'streamflow' in the text.

---

## Author Comment (AC2)

This manuscript presents an interesting study on root zone storage capacities. However, I have some rather fundamental concerns, namely, using ERA5 data and neglecting parameter uncertainties. Both are straightforward to address, but this would require significant work for new model runs. Below, I discuss these two concerns and list a few minor concerns:

We thank Anonymous Referee # 2 for supporting the relevance of our study. Please find our detailed responses to your concerns highlighted in blue.

Major comments

Use of ERA5 data: there might be good reasons to use ERA5 data. These data are, for instance, a suitable option for global studies. However, one has to be aware of the substantial uncertainties of these data and for the US, the use of 'real' (station-based) observations is preferable, especiallyy for precipitation data (for a detailed analysis see https://egusphere.copernicus.org/preprints/2024/egusphere-2024-864/). From the text, I am not fully sure whether ERA5 data is used for both approaches or only the second one (MCT). Still, both cases would be problematic and I would recommend using station-based data, especially for precipitation, in both cases.

Reply: We thank the referee for their careful review and constructive feedback. We acknowledge the uncertainties associated with ERA5 reanalysis data and their suggestion to consider station observation data instead. In our study, we adopted two distinct approaches to estimate root zone storage capacity. The first method employed a model calibration technique using station observed precipitation and CAMELS streamflow data. The second method utilized MCT with ERA5 data, a leading reanalysis dataset known for its comprehensive spatial and temporal coverage, thereby providing an independent validation of our results.
Since the precipitation, evaporation, and snowmelt data in ERA-5 are all in the same spatial and temporal resolution, if we use ERA-5 evaporation and Caravan precipitation as the Anonymous Referee #2 suggested, it brings in spatial mismatch, which could result in even larger uncertainty. Therefore, we propose to continue utilizing ERA5 for both evaporation and precipitation data throughout our analysis.

While details on how the model has been calibrated are missing, my understanding from reading the manuscript is, that there was one single parameter set for each catchment. This would not be state-of-the-art but one should consider an ensemble of good/acceptable parameterisations. Please explain how parameter uncertainty was considered. If it was neglected, as I am afraid of, I would recommend to use some approach that allows for considering multiple parameter sets.

Reply: We will provide a clearer description of the parameter calibration process and the parameter uncertainties in the manuscript. Specifically, in our study, each catchment underwent extensive parameter calibration where we generated 40,000 different parameter combinations randomly. These combinations were evaluated using the Kling-Gupta Efficiency (KGE) coefficient, and the top 1% performing combinations (equating to 400 parameter sets)

were selected. We then averaged these top sets to derive the final parameter set for each of the 497 study catchments. This approach comprehensively addresses parameter uncertainty across our analysis.

Minor comments:

L122: please provide details on how the model was calibrated

Reply: Section 2.2.2 outlines the model calibration process in detail. In the revision, we will enrich this section with additional specifics and clarity regarding the calibration methodology.

L153: this sounds like the soil routine in the HBV model. If yes, a reference to Bergström et al. would be suitable.

Reply: The root zone module constitutes the central component of the FLEX model. In our discussion manuscript, we acknowledged the HBV soil routine and referenced pertinent literature (Lindström et al., 1997; Seibert and Bergström, 2022). However, for this study, we opted to utilize the beta function from the Xinanjiang model (Zhao, 1992) instead of the HBV parameterization.

L169ff: with the focus on soil water storage, I wonder whether it is appropriate to use KGE as the objective function, which is known to focus on performances during high-flow events.

Reply: Firstly, the efficiency of objective functions falls outside the scope of this study. Previous research has extensively discussed this issue (Fenicia et al., 2007; Gupta et al., 2009). In our study, we employed the Kling-Gupta Efficiency (KGE) as our objective function, which is commonly used in hydrological modeling evaluations. KGE assesses the performance across all range of flows, including high-flow, moderate-flow, and low-flow conditions.

L182: Sorry, but I am not sure I fully understand how parameter sets are combined within one cluster. Despite the grouping by Jehn, parameter sets must vary largely within each cluster. If anything, aggregating simulated time series might be more appropriate than averaging parameter values.

Reply: Thank you for bringing this to our attention. To clarify, we did not combine parameter sets within a cluster. Initially, we calibrated parameters individually for each catchment. Subsequently, we computed the average parameter set for all catchments within a cluster. We agree with the reviewer that despite clustering, there are substantial variations in precipitation-runoff dynamics among different catchments. We will ensure to articulate this more clearly in the revised manuscript.

Equations: please avoid multi-letter variable names, these are mathematically incorrect (SM

means S times M)

Reply: We will modify these variable names in the revision.

Figures: I find the figures hard to read. Partly this might be due to the small size. In Figure 5 the different axes use a different scale, which is unfortunate.

Reply: In the revised version, we will redraw these figures. The varying scales used in Figure 5 are designed to effectively illustrate the diverse range of root zone storage capacities.

References
Zhao, Ren-Jun. (1992). The Xinanjiang model applied in China. Journal of Hydrology, 135(1–4), 371–381. https://doi.org/10.1016/0022-1694(92)90096-E
Lindström, G., Johansson, B., Persson, M., Gardelin, M., & Bergström, S. (1997). Development and test of the distributed HBV-96 hydrological model. Journal of Hydrology, 201(1–4), 272–288. https://doi.org/10.1016/S0022-1694(97)00041-3
Fenicia, F., H. H. G. Savenije, P. Matgen, and L. Pfister (2007), A comparison of alternative multiobjective calibration strategies for hydrological modeling, Water Resour. Res., 43, W03434, doi:10.1029/2006WR005098.
Gupta, H. V., Kling, H., Yilmaz, K. K., & Martinez, G. F. (2009). Decomposition of the mean squared error and NSE performance criteria: Implications for improving hydrological modelling. Journal of Hydrology, 377(1–2), 80–91.
Seibert, J., & Bergström, S. (2022). A retrospective on hydrological catchment modelling based on half a century with the HBV model. Hydrology and Earth System Sciences, 26(5), 1371–1388. https://doi.org/10.5194/hess-26-1371-2022